# MTA Enhances the Potential of Adipose-Derived Mesenchymal Stem Cells for Dentin–Pulp Complex Regeneration

**DOI:** 10.3390/ma13245712

**Published:** 2020-12-15

**Authors:** Danial Babaki, Kagya Amoako, Ahmad Reza Bahrami, Sanam Yaghoubi, Mahdi Mirahmadi, Maryam M. Matin

**Affiliations:** 1Department of Biomedical Engineering, Tagliatela College of Engineering, University of New Haven, West Haven, CT 06516, USA; dbaba2@unh.newhaven.edu (D.B.); kamoako@newhaven.edu (K.A.); 2Department of Biology, Faculty of Science, Ferdowsi University of Mashhad, Mashhad 9177948974, Iran; ar-bahrami@um.ac.ir; 3Industrial Biotechnology Research Group, Institute of Biotechnology, Ferdowsi University of Mashhad, Mashhad 9177948974, Iran; 4Center for Cancer Research, National Cancer Institute, NIH, Bethesda, MD 20892, USA; sanam.yaghoubi@nih.edu; 5Stem Cells and Regenerative Medicine Research Department, Iranian Academic Center for Education, Culture and Research (ACECR), Mashhad Branch, Mashhad 9177948974, Iran; mahdi.mirahmadi85@gmail.com; 6Novel Diagnostics and Therapeutics Research Group, Institute of Biotechnology, Ferdowsi University of Mashhad, Mashhad 9177948974, Iran

**Keywords:** mesenchymal stem cells, osteogenic differentiation, odontogenesis, mineral trioxide aggregate

## Abstract

The aim of the current study was to investigate the effects of mineral trioxide aggregate (MTA) on the proliferation and differentiation of human adipose-derived mesenchymal stem cells (Ad-MSCs) as a surrogate cell source in futuristic stem-cell-based endodontic therapies. Human Ad-MSCs and mesenchymal stem cells derived from bone marrow (BM-MSCs) were isolated from liposuction waste adipose tissue and femur, respectively, and the effects of MTA-conditioned media on their viability, mineralization potential, and osteo/odontogenic differentiation capacity were subsequently evaluated. Alkaline phosphatase (ALP) activity, quantitative alizarin red S staining, and quantitative reverse transcription-polymerase chain reaction (qRT-PCR) analyses were performed to investigate and compare the osteo/odontogenic induction potential of MTA on the Ad/BM-MSCs. The results of cytotoxicity assay revealed that at different concentrations, MTA-conditioned medium was not only biocompatible toward both cell types, but also capable of promoting cell proliferation. ALP activity assay showed that 0.2 mg/mL was the optimal concentration of MTA-conditioned medium for osteo/odontogenic induction in Ad/BM-MSCs. The expression of osteo/odontogenic gene markers was increased in Ad/BM-MSCs treated with 0.2 mg/mL MTA-conditioned media. Our results indicated that MTA can efficiently enhance the osteo/odontogenic potential of Ad-MSCs, and thus they can be considered as a better cell source for dentin–pulp complex regeneration. However, further investigations are required to test these potentials in animal models.

## 1. Introduction

For more than a decade, use of stem cells and biomaterial-based approaches has been a focus for endodontic therapy [1]. Although contemporary root canal treatments have overcome the drawbacks of conventional approaches, they cannot provide an equivalent structure comparable to dentin–pulp complex in terms of morphology and function [2,3]. So, the concept of dentin–pulp regeneration using mesenchymal stem cells (MSCs) combined with scaffold and/or biomaterials aims to improve the dental therapeutic procedures. In particular, a three-dimensional culture of MSCs at a specific density can potentially generate a dentin/pulp-like structure both in vitro and in vivo. Such bioengineered tissues would not only enhance the perceptive potential in response to pernicious stimuli but also result in a functionally dynamic tissue [4,5]. Massive attempts throughout the world have led to great progress in this field. However, many questions remain unsolved concerning the candidate cell sources, the favorable types of biomaterials or scaffolds, and the ideal means of differentiation [6].

Odontoblasts are of fundamental importance in the structure and function of both dentin and pulp. These cells are mostly located at the dentin–pulp border zone and play a crucial role in secreting the dentin organic matrix and its mineralization, and for this reason, there are many studies focusing on the stem cell differentiation towards odontoblast-like cells [6,7]. Researchers have been trying to address the optimal source of MSCs and the appropriate differentiation procedure, more specifically including chemical treatments and genetic manipulations. The results from in vitro studies suggest that the odontoblastic differentiation potential of MSCs derived from bone marrow and dental tissue can potentially improve the structure and the functionality of dentin–pulp complex engineered constructs. These have been confirmed by gene and protein expression assays, mineralization assessment, and migration assay [6,8,9].

Many cases of dentin–pulp lesions are associated with internal resorption of the root, root perforation, and open apex [10,11,12,13]. So, the involvement of dental materials in the future of stem-cell-based therapies targeting the dentin pulp degeneration is inevitable. In other words, the bioactivity of the dental material should be considered and manipulated in favor of stem cell therapeutic approaches, when an implanted bioengineered construct for replacing the dentin–pulp complex is in direct contact with the dental material used for restoring the hard tissue [4,14,15]. In addition to proper chemical and mechanical characteristics, an optimal material for dental tissue bioengineering should bear antibacterial and antifungal properties, biocompatibility, and bioactivity [6,7]. Among root canal filling materials, investigators have widely reported the successful application of mineral trioxide aggregate (MTA) in vital pulp therapy [16]. Moreover, bioactivity and cytotoxicity measurements revealed that MTA not only shows biocompatibility toward bone marrow and dental-derived mesenchymal stem cells but also can induce their proliferation and osteo/odontoblast-like differentiation in a concentration-dependent manner [6,17].

MSC isolation from adipose tissue has circumvented many concerns associated with application of bone marrow and dental tissue as other stem cell sources [18]. In addition to the fact that the tissue is more accessible, replenishable, and abundant, stem cells can be easily isolated from liposuction waste tissue by enzymatic digestion and centrifugation. It is worth mentioning that liposuction has a lower surgery risk, and patients experience minimal post operation discomfort. Furthermore, evidence from numerous in vitro and in vivo evaluations of therapeutic efficacy of adipose-derived mesenchymal stem cells (Ad-MSCs) in combination with cytokines and biomaterials demonstrated that Ad-MSCs have highly consistent immunotypes and multilineage differentiation potential [18,19,20,21]. The aim of the present study was to compare the effects of MTA on the proliferation and osteo/odontogenic differentiation potential of mesenchymal stem cells derived from bone marrow (BM-MSCs) and adipose tissue.

## 2. Materials and Methods

### 2.1. Isolation and Culture of Human Ad-MSCs and BM-MSCs

Informed consents were signed by each donor prior to the surgical procedures, and the research was approved by the ethics committee at Ferdowsi University of Mashhad (IR.UM.REC.1399.079). Ad-MSCs were isolated and characterized, using the liposuction waste adipose tissue as previously reported [22]. Dulbecco’s modified Eagle’s medium-low glucose (DMEM-LG, Thermo Fisher Scientific, Waltham, MA, USA) supplemented with 10% fetal bovine serum (FBS, Thermo Fisher Scientific, Waltham, MA, USA), 100 U/mL penicillin, and 100 μg/mL streptomycin (Thermo Fisher Scientific, Waltham, MA, USA) was used for culturing the Ad-MSCs. Cell cultures were maintained at 37 °C in a humidified 95% air/5% CO_2_ atmosphere, with a change of medium every 2–3 days. BM-MSCs were obtained following bone marrow (BM) aspiration from the femur of patients undergoing orthopedic surgery. The samples were maintained for 3 days in the 75-cm^2^ cell culture flasks containing 20 mL of DMEM-LG supplemented with 10% FBS and penicillin–streptomycin. Subsequently, cultures were rinsed with phosphate-buffered saline (PBS) to remove any non-adherent cells, and then fresh media were added to the flasks. The maintenance condition was similar to that of Ad-MSCs. The identity of both cell types was confirmed by evaluating the cell surface marker expression profile, adipo/osteogenic differentiation potentials, and cell morphology analysis as previously described [22]. MSCs from passage 4 were used for the experiments. 

### 2.2. Preparation of MTA-Conditioned Medium

According to the manufacturer’s instructions, MTA (ProRoot; Dentsply, Tulsa Dental, Tulsa, OK, USA) was mixed with sterile water in a laminar flow hood under aseptic conditions. The mixture was incubated at room temperature for 24 h to set completely and then ground to a fine powder. Subsequently, by using a 45 μm strainer, 200 mg of the powder was filtered and mixed with 1 mL of DMEM-LG. Then, the bioactive ingredients of MTA were released into the medium through the incubation of the suspension at 37 °C in a humidified 95% air/5% CO_2_ atmosphere. After 1 week, the filtered supernatant (through a 2.5-μm syringe filter) was referred to as “MTA-conditioned medium” at the concentration of 200 mg/mL. Different concentrations of MTA-conditioned media (0.02, 0.2, 2, and 20 mg/mL) were then prepared by dilution of the 200 mg/mL solution with culture media. For the viability and differentiation assays, cells were treated every other day with MTA-conditioned media. Unconditioned DMEM-LG supplemented with 10% FBS and antibiotics was used for the negative control group in all experiments. For both experimental and control groups, the following cell viability and differentiation-inducing assays were performed in triplicate.

### 2.3. Cell Viability Assay

3-(4,5-Dimethylthiazol-2-yl)-2,5-diphenyl tetrazolium bromide (MTT)-based cell viability assay was performed to investigate the effects of MTA on proliferation of MSCs. For this purpose, cells were seeded in 96-well plates at a concentration of 4 × 10^3^ cells per well in 200 μL culture medium. Then, subsequent to 24-h starving incubation in the serum-free media, cells were treated with various concentrations of MTA-conditioned media. After 3, 5, and 7 days, 20 μL MTT solution (5 mg/mL; Tinab Shimi Khavaremiyaneh Co., Mashhad, Iran) was added to each well and incubated at 37 °C for 4 h. Next, the culture media were removed, and the formazan crystals were solubilized by adding 150 μL dimethyl sulfoxide (DMSO) to each well. Finally, the optical density values were measured at 545 nm using a microplate reader (Awareness Technology Inc., Palm City, FL, USA). 

### 2.4. Alkaline Phosphatase (ALP) Activity and Alizarin Red S Staining

The ALP assay was performed to identify the optimal concentration of the MTA-conditioned medium. Briefly, MSCs (4 × 10^3^ cells per well) seeded in 96-well plates were treated with various concentrations of MTA-conditioned media. After 14 days, ALP activity of cell lysates was determined using the p-nitrophenyl phosphate hydrolysis method and measuring the p-nitrophenyl phosphate concentration by an ELISA (enzyme-linked immunosorbent assay) reader at 405 nm (Awareness Technology Inc., Palm City, FL, USA). Next, using Bradford method, the total protein concentration of each sample was measured, and according to that, the ALP activity was normalized and reported as nmol/μg protein/min. Comparing to control groups, at the concentration of 0.2 mg/mL, both Ad-MSCs and BM-MSCs showed significantly higher ALP activity. So, this concentration was considered for the following differentiation-inducing experiments.

In order to analyze the mineralization-inducing potential of MTA, MSCs were seeded in 6-well plates and treated with 0.2 mg/mL MTA-conditioned media for 14 days. Then, cells were fixed with 4% paraformaldehyde (Sigma-Aldrich, St. Louis, MO, USA) for 30 min, washed with deionized H_2_O, and incubated in 2 mL alizarin red S (ARS, 40 mmol/L, pH = 4.1, Sigma-Aldrich, St. Louis, MO, USA) for 20 min at room temperature. The cells were then rinsed with deionized H_2_O, and finally, the mineralized nodules were studied under an inverted microscope (Nikon, Tokyo, Japan) for qualitative evaluation. As previously reported, to semiquantify the mineralization rate, the absorbed dye was extracted and the optical density values for different samples were measured at 405 nm [23]. Briefly, subsequent to incubation of stained calcium deposits at −20 °C for 24 h, 800 μL of 10% (*v*/*v*) acetic acid was added to each well, and the plate was incubated at room temperature for 30 min. Next, the cell monolayer was detached using a cell scraper, and the mixture was incubated at 85 °C for 10 min. The slurry mixture was then centrifuged at 20,000× *g* and 200 μL of the supernatant was transferred to each well of a 96-well plate containing 80 μL 10% (*v*/*v*) ammonium hydroxide. Finally, the optical density values were measured at 405 nm using a microplate reader (Awareness Technology Inc., Palm City, FL, USA).

### 2.5. Evaluation of Gene Expression by Quantitative Reverse Transcription-Polymerase Chain Reaction (qRT-PCR)

To determine the effect of MTA on the osteo/odontogenic markers, the expression of ALP, Runt-related transcription factor 2 (*RUNX2*), osteocalcin (*OCN*), and dentin sialophosphoprotein (*DSPP*) was analyzed quantitatively. To do so, total RNA was extracted using the TREK Kit (Riz Molecule DANA, Mashhad, Iran). RNA concentration and purity, determined by 260/280 and 260/230 ratios, were measured with a Nanodrop ND-1000 spectrophotometer (Nanodrop Technologies Inc., Wilmington, DE, USA). The extracted RNAs were used as template in the standard reverse-transcription reaction to make cDNA, using RevertAid First Strand cDNA Synthesis Kit (Thermo Fisher Scientific Inc., MA, USA). Quantitative RT-PCR was performed by CFX96 Touch Real-Time PCR Detection System (Bio-Rad, Hercules, CA, USA), using SYBR Green Real-Time Master Mix (Takara Bio, Kyoto, Japan). The final amplification volume was 17.5 μL. The specific primers are listed in Table 1. Glyceraldehyde 3-phosphate dehydrogenase (*GAPDH*) was used as an internal control and each reaction was performed in triplicate. The relative gene expression was calculated using the negative control group as the calibrator.

### 2.6. Statistical Analysis

All results are expressed as mean ± SD (standard deviation). Statistical analysis was performed by GraphPad Prism v6.07 software. Firstly, the Shapiro–Wilk test was used to evaluate the normality of data sets acquired from each experiment. Comparisons between two independent groups were made by Student’s *t*-test and *p* values less than 0.05 were considered to be statistically significant.

## 3. Results

### 3.1. Effects of MTA on Proliferation of Ad-MSCs and BM-MSCs

Flow cytometry analyses confirmed the characteristics of both mesenchymal stem cell types used in this study. While they were negative for CD11b and CD45, typical surface markers of MSCs (CD73, CD90, and CD105) were shown to be expressed by the cells (Figure 1).

Results from MTT assay showed that at various time points, MTA-conditioned media not only had no toxicity at different concentrations but also could stimulate cell proliferation at the concentrations of 0.2 and 2 mg/mL, after 7 days of treatment (Figure 2). However, there was no significant difference in proliferation-inducing potential of MTA between MSCs derived from adipose tissue and those derived from bone marrow.

### 3.2. Effects of MTA on Osteo/Odontogenic Potential of Ad- and BM-MSCs

At different time points, Ad- and BM-MSCs treated with 0.2 mg/mL MTA-conditioned media had significantly higher ALP activity compared to the corresponding control groups, while there was no significant difference between ALP activity levels among other concentrations and control groups (Figure 3). Thus, for investigating the differentiation-inducing potential of MTA, a concentration of 0.2 mg/mL MTA was considered in the following experiments.

After 14 days, alizarin red S staining indicated that MTA-treated MSCs presented a notable increase in the number of mineralized nodules compared with control groups (Figure 4). 

Interestingly, the semiquantification of ARS in a stained monolayer by acetic acid extraction indicated that extracellular mineralization was significantly higher in MTA-treated Ad-MSCs compared with BM-MSCs (Figure 5). 

The purity of extracted total RNA used for reverse transcription was good, with 260/280 ratios about 2 and 260/230 ratios about 2.1 in all samples. Using the geNorm software (version 3.5) in which the suggested cut-off M-value is 1.5, the M-value for expression of the housekeeping gene was calculated as 1.21, indicating the invariant expression of *GAPDH* under the experimental conditions in different groups. Figure 6 shows the relative gene expression after 14 days upon the exposure of cells to 0.2 mg/mL MTA-conditioned media. After 14 days, MTA-conditioned media enhanced the expression of all osteo/odontogenic gene markers in both cell types in comparison to corresponding untreated groups. Among cells exposed to MTA extract, the expression levels of *RUNX2* and *DSPP* were found to be higher in the Ad-MSC group compared to the BM-MSC group. No significant difference was detected in the expression of other osteo/odontogenic gene markers among treated MSCs (Figure 6).

## 4. Discussion

The goal of conventional regenerative endodontic procedures, such as apexogenesis, is to regenerate the dentin–pulp complex using biomaterials and cells, specifically dental stem cells, at the lesion site [24,25]. To date, several studies have evaluated the bioactivity and biocompatibility of dental biomaterials towards mesenchymal stem cells isolated from the pulp tissue and the periodontium [26,27]. Because of its suitable mechanical properties, appropriate biocompatibility, bioactivity, and better clinical outcome, the significance of MTA was underscored in such reports [16,28,29,30]. According to these studies, MTA was not only a biocompatible material toward BM-MSCs but also, in some cases, it was capable of enhancing cell growth and proliferation, whereas it could be toxic for tooth-resident MSCs in a time- and concentration-dependent manner. Moreover, the results from gene and protein profiling assays showed that MTA can stimulate or enhance the expression of osteo/odontogenic markers. These markers particularly include *RUNX2*, *DSPP*, *OCN*, and ALP. Therefore, based on these promising applications in the clinic and in vitro studies focusing on stem cell response to MTA treatment, it is believed that this biomaterial could play a crucial role in future stem-cell-based therapies for dentin–pulp complex.

On one hand, in accordance with the results from previous research studying the relative superiorities of Ad-MSCs compared to the MSCs derived from bone marrow and dental tissues, for the first time in the current study, the biocompatibility and the osteo/odontogenic induction potential of MTA on Ad-MSCs were evaluated and compared to those on BM-MSCs, in the hope that Ad-MSCs can be used as a promising surrogate cell source for therapeutic purposes in endodontic treatments. On the other hand, there are other reports indicating that the tooth-resident stem cells are better options for this purpose. Considering these results, further studies should compare the osteo/odontogenic potential of these cells with the ones that were investigated in this experiment [31].

Similar to previous reports, MTT assay indicated that MTA at different concentrations not only shows biocompatibility toward Ad-MSCs but also can induce their proliferation (Figure 2). These results also demonstrated that there was no statistically significant difference in proliferating potential of Ad- and BM-MSCs following MTA treatment. Moreover, our results revealed that MTA-treated human Ad/BM-MSCs presented higher differentiation potential toward osteo/odontoblast-like cells than the untreated cells, as indicated by increased ALP activity, upregulated formation of mineral deposits, and enhanced expression of osteo/odontogenic gene markers (Figure 3, Figure 4, Figure 5 and Figure 6). Interestingly, it was shown that MTA-treated Ad-MSCs had more capacity for mineralized nodule formation than BM-MSCs. Furthermore, the increase rate in the expression of *RUNX2* and *DSPP* was statistically higher in Ad-MSCs than that of BM-MSCs following treatment with MTA-conditioned media. *RUNX2* and *DSPP* are key master regulators associated with early stages of osteo/odontoblast differentiation. At later stages, downstream regulators, such as *OSX* and *OCN*, function to implement the final steps of the nucleation phase in dentin calcification and bone development. As previously demonstrated, the mitogen-activated protein kinase (MAPK) and nuclear factor kappa B (NF-κB) pathways are involved in the overexpression of these master regulators and downstream elements, and play an important role in osteo/odontogenic induction potential of MTA on both BM-MSCs and dental stem cells [32,33]. Particularly, it has been shown that MTA-conditioned media upregulate the MAPK, leading to the enhancement of *RUNX2* expression. Moreover, inhibiting the NF-κB pathway can not only inhibit the mesenchymal stem cells’ proliferation but also reduces the osteo/odontogenic potential of these cells when they are in direct contact with MTA. All in all, the results of this and other contemporary studies suggest that in the concept of stem-cell-based dentin–pulp complex regeneration, MTA can serve as a proper dental material for hard tissue restoration, and can induce proliferation and differentiation of stem cells within the implanted bioengineered construct [32,33]. On top of that, by gene editing, the expression of master regulators involved in the proliferation and differentiation can be modified in favor of this purpose. 

There are many studies investigating and introducing different types of stem cells as the candidate cell source for regenerative medicine. Presently, the advantages associated with the use of mesenchymal stem cells over induced pluripotent stem cells (iPSCs) are rooted in the iPSC problems of production efficiency, and application safety in regenerative medicine [34,35]. Consequently, in the current investigation, two types of mesenchymal stem cells were utilized, and it was concluded that Ad-MSCs can be considered as a better cell source when compared to BM-MSCs.

## 5. Conclusions

Although more in vitro/in vivo studies are required to investigate mechanisms embedded in MTA-mediated osteo/odontogenic differentiation of Ad-MSCs, it has been shown in the current study that 0.2 mg/mL MTA-conditioned medium can induce proliferation and promote differentiation of Ad-MSCs towards osteo/odontoblast-like cells. These findings can greatly improve the application of MTA in tooth regenerative medicine.

## Figures and Tables

**Figure 1 materials-13-05712-f001:**
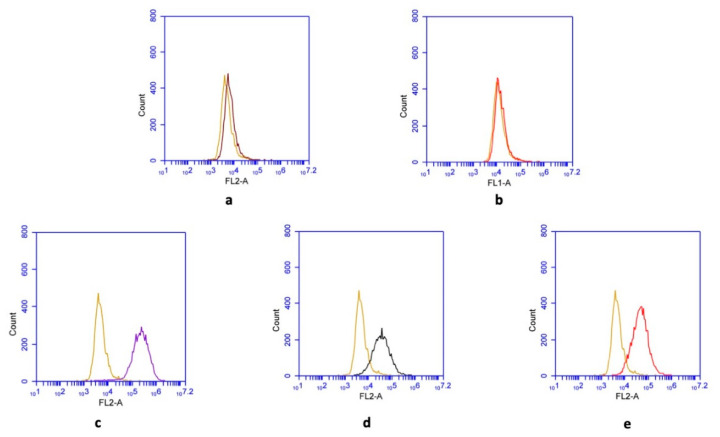
Cell surface marker expression profile of mesenchymal stem cells (MSCs). Flow cytometric analysis demonstrating that obtained MSCs were negative for (**a**) CD11b and (**b**) CD45, while they were positive for the antigens (**c**) CD73, (**d**) CD90, and (**e**) CD105. Only representative examples are shown here.

**Figure 2 materials-13-05712-f002:**
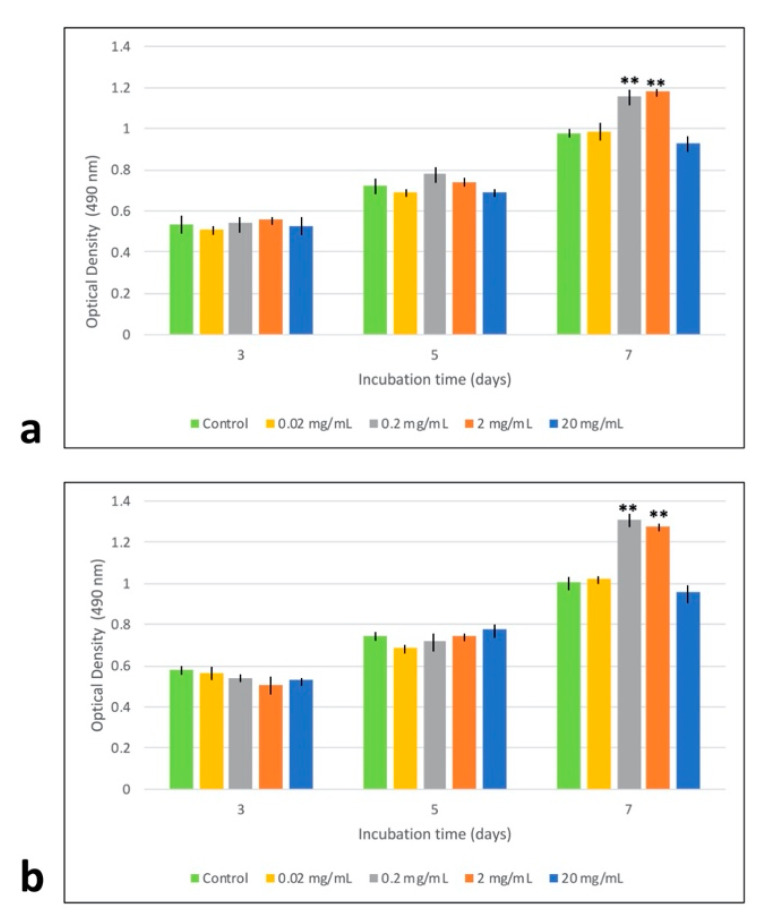
Investigating the effects of mineral trioxide aggregate (MTA)-conditioned media on the viability of human MSCs by MTT assay. Viability of (**a**) mesenchymal stem cells derived from bone marrow (BM-MSCs) and (**b**) human adipose-derived mesenchymal stem cells (Ad-MSCs) following treatment with different concentrations of MTA extract after 3, 5, and 7 days. At different time intervals, various concentrations of MTA-conditioned media had no cytotoxic effects on Ad/BM-MSCs. Interestingly, 7 days of incubation in 0.2 and 2 mg/mL MTA-conditioned media could induce proliferation in both cell types. Mean ± SD, n = 3, ** indicates *p* < 0.01.

**Figure 3 materials-13-05712-f003:**
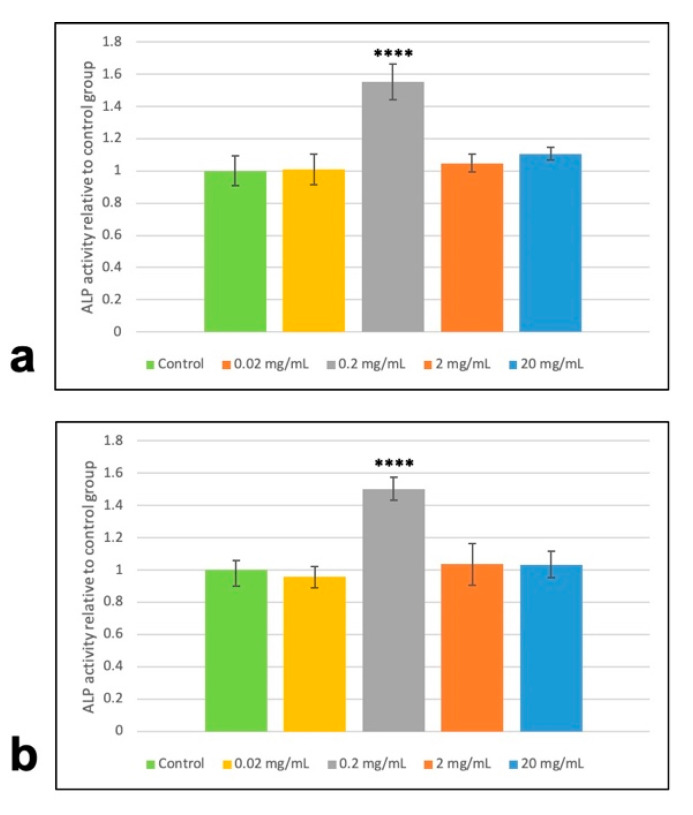
Investigating the effects of MTA-conditioned media on alkaline phosphatase activity (ALP) of human MSCs. ALP activity of (**a**) BM-MSCs and (**b**) Ad-MSCs following treatment with different concentrations of MTA-conditioned media after 14 days. Human MSCs incubated in 0.2 mg/mL MTA-conditioned media had significantly higher ALP activity compared to the control and other treated groups. Values are mean ± SD, n = 3, **** indicates *p* < 0.0001.

**Figure 4 materials-13-05712-f004:**
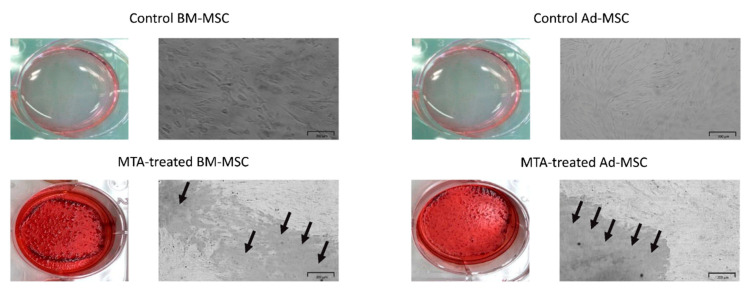
Alizarin red S staining of Ad/BM-MSCs after 14 days of treatment with 0.2 mg/mL MTA-conditioned media. No mineralized nodule formation can be seen in control groups. The arrows indicate the mineral deposits prior to dye extraction for semiquantification. Scale bar represents 200 μm.

**Figure 5 materials-13-05712-f005:**
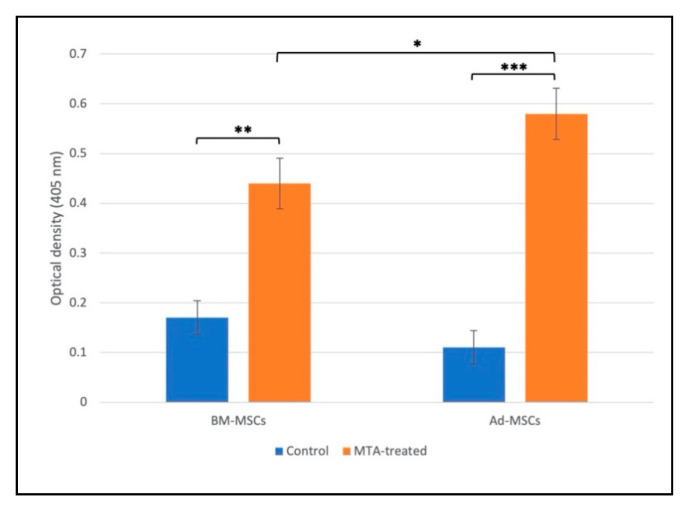
Alizarin red S (ARS) acid extraction to semiquantify the production of mineral deposits. After 14 days of incubation in 0.2 mg/mL MTA-conditioned media, Ad/BM-MSCs exhibited significantly higher amounts of absorbed dye than control groups. Furthermore, semiquantitative analysis of ARS staining revealed that the Ad-MSCs treated with MTA-conditioned medium produced significantly more stained mineral deposits than BM-MSCs. Values are mean ± SD, n = 6; *, **, and *** indicate *p* values less than 0.05, 0.01, and 0.001, respectively.

**Figure 6 materials-13-05712-f006:**
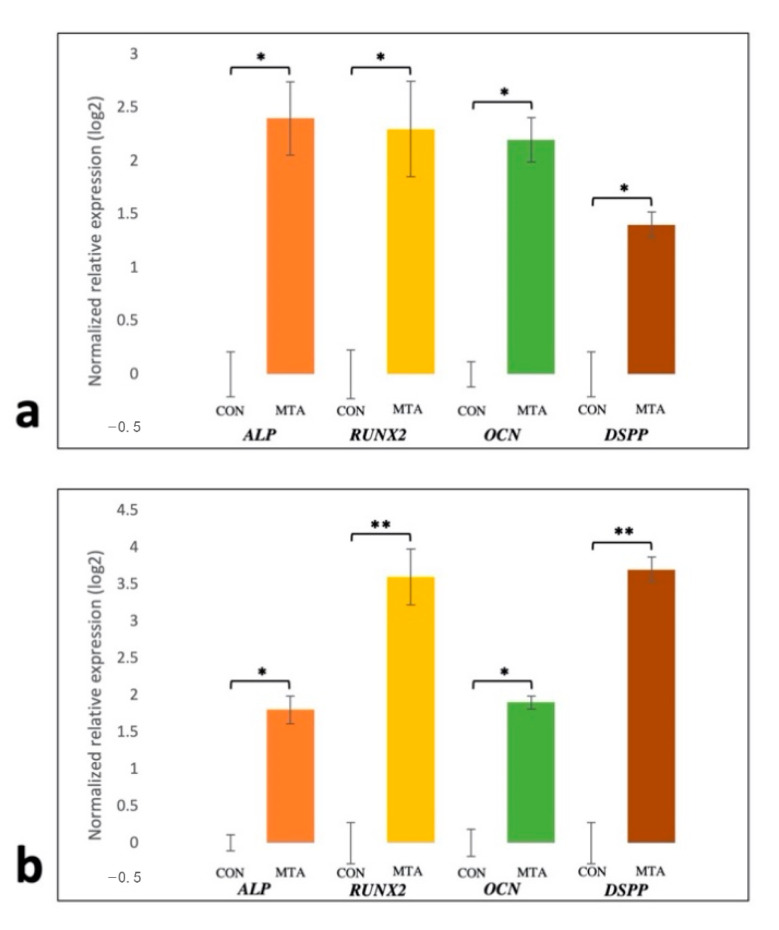
Expression of osteo/odontogenic gene markers (ALP, *RUNX2*, *OCN*, and *DSPP*) in human MSCs after 14 days of incubation in 0.2 mg/mL MTA-conditioned media. All gene markers were significantly upregulated compared to the corresponding control group. The rate of increase in the expression of *RUNX2* and *DSPP* was higher in Ad-MSCs (**b**) than BM-MSCs (**a**). *GAPDH* was used as an internal control. Values are mean ± SD, n = 3; * and ** indicate *p* values less than 0.05 and 0.01, respectively.

**Table 1 materials-13-05712-t001:** Sense and antisense primers used for qRT-PCR.

Gene	Primer	Sequence (5′→3′)	Product Length
ALP	Forward	ACCAAGCGCAAGAGACACTG	106 bp
	Reverse	GTGGAGACACCCATCCCATCT	
*DSPP*	Forward	CAAAAGTCCAGGACAGTGGGC	186 bp
	Reverse	TGGTTTGCTTTGAGGAACTGGA	
*OCN*	Forward	CACCGAGACACCATGAGAGC	132 bp
	Reverse	CTGCTTGGACACAAAGGCTGC	
*RUNX2*	Forward	CTGTCATGGCGGGTAACGAT	132 bp
	Reverse	AGGTGAAACTCTTGCCTCGT	
*GAPDH*	Forward	GTCGGAGTCAACGGATTTGG	156 bp
	Reverse	ATGGAATTTGCCATGGGTGG	

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
