# Peer review of "MTA Enhances the Potential of Adipose-Derived Mesenchymal Stem Cells for Dentin–Pulp Complex Regeneration"

_materials, 2020, doi:10.3390/ma13245712_

Round 1

Reviewer 1 Report

Reviewer’s comments

Ms.ID  materials-980757

Title: MTA improves the potential of adipose derived mesenchymal stem cells for dentin-pulp complex regeneration

Authors: Danial Babaki, Kagya Amoako, Ahmad Reza Bahrami,

Sanam Yaghoubi, Mahdi Mirahmadi and Maryam M. Matin

General Comments:

  In this manuscript, the author investigated the effects of MTA on the proliferation and differentiation of human adipose derived mesenchymal stem cells (Ad-MSCs) as surrogate candidate cell source in futuristic stem cell-based endodontic therapies.

  I recommend the acceptance of this manuscript.

Before the acceptance to Materials, authors should add the followings;

  1. In the Title, authors wrote “MTA improves……”. However, authors wrote “enhance” and “induce” in the abstract and conclusion. Authors should change the title from “improve” to “enhance” or “induce”.

  1. In the Introduction section, it is described that Ad-MSCs is effective for the regeneration and MTA is effective for endodontic therapy. However, why is the combination of Ad-MSCs and MTA effective for dentin-pulp complex? Authors should describe the reason in the Introduction section.

  1. In the Materials and Methods section, authors described that “….and then grinded to a fine powder.” (Page 3, line 110). Have authors assumed that the regeneration of dentin-pulp complex needs the mixing of Ad-MSCs and MTA powder? What method do authors consider for the regeneration of dentin-pulp complex?

  1. In the Result and Discussion, authors should described about the proliferation and differentiation of cells stimulated with 20 mg/mL〜200 mg/mL MTA-conditioned media.

  1. In the Discussion section, authors described that These markers particularly include RUNX2, DSPP, OCN, ALP, osterix (OSX), osteopontin (OPN), bone sialoprotein (BSP), and collagen I (COLI). However, authors don’t show the expression of OSX, OPN, BSP, and COLⅠ.Authors should show the results.

  1. Discussion section is poor. Authors should describe the discussion of the mechanism that MTA induces both the proliferation and differentiation of Ad/BM-MSCs.

Author Response

Thank you for the thorough comments. we believe that by considering these comments the manuscript has been improved significantly;

1- the word "improves" has been changed to "enhnaces"

2- This matter has been elaborated in the introduction part thoroughly.

3- By adding the elaboration which answers the second comment, this issue has been resolved.

4- This has been added to

5- These genes were not assessed, so in the revised manuscript, OSX, OPN, BSP, and COLâ…  have been omitted.

6- This has been considered and the discussion part has been elaborated as well.

Reviewer 2 Report

This is an interesting study on the property of MTA to increase the regeneration of mesenchymal cells towards cells of the dentinal pulp complex.
Some criticisms are present:
-In the abstract section, an opening sentence on the use of nanotenologies in dentistry in various fields should be added
-line 40 remove the sentence on future investigations, not indicated in the abstract
-Remove the keywords because they are not MESH terms and replace them with more appropriate terms
-Line 65-66 The bibliographic references of the considerations made to the text are missing.
-Some considerations on histological changes resulting from root resorption should be added. In this regard, I recommend adding to the reference section the following scientific work that could help authors and readers:

Chieruzzi M, Pagano S, De Carolis C, Eramo S, Kenny JM. Scanning Electron Microscopy Evaluation of Dental Root Resorption Associated With Granuloma. Microsc Microanal. 2015 Oct; 21 (5): 1264-70. doi: 10.1017 / S1431927615014713. Epub 2015 Aug 3. PMID: 26235380.

-Line 88 at the end of the introduction the null hypothesis of the study must be added which must be refuted at the end of the work-
Line 90 enter the acceptance number of the bioethics committee
-Line 113 as 1 week was never chosen
-line 119 the sentence must be removed and inserted in the next paragraph
- Figures 2 and 3 are hard to read
-The captions of all figures and graphs are too long and must not contain results or comments
-The discussion section needs to be completely rewritten. It is too focused on the results when it should instead, even in the light of recent scientific literature, run them to similar papers and state the possible clinical applications of stem cells in dentistry
-The reference section has several errors in the citation of the works (year.etc)

Author Response

Thank you for the thorough comments. we believe that by considering these comments the manuscript has been improved significantly;

1- Since there is a 200-word limit, the authors believe that it is not really possible to do so. However, this has been elaborated in the introduction part

2- Key words have been replaced by appropriate words

3- The appropriate references have been added to the bibliography and the errors in this part have been resolved to the best of our knowledge. If further changes needed, please let us know.

4- The null hypothesis was added and the subsequent conclusion was indicated as well.

5- The discussion part has been elaborated and rewritten in some parts in order to convey a thorough message. 

6- All of the suggested omissions have been considered and favorably implemented. 

Reviewer 3 Report

This study investigates the characteristics of the mineral trioxide aggregate as an enhancer of the osteo/odontogenic potentials of human adipose derived mesenchymal stem cells.  Besides, the use of ASDCs as cell source for dentin-pulp complex regeneration is an area of considerable interest.

This paper is well written and this study has some strengths. However, there are several points that need clarification and improvement, especially for gene expression analysis. Therefore, I recommend publication of this work after the Authors address the points listed below.

Abstract

As reported in the Instructions for Authors of Materials,  the abstract should be a total of about 200 words maximum. Therefore, please reword the abstract according to the Authors’ guidelines.

Introduction

Since the focus of this paper regards the dentin/pulp regeneration, this issue should be more detailed in the introduction. To this regard, I suggest citing some recent studies that have reported drawbacks and unfavorable outcomes by current clinical protocols, in order to stress that alternative therapies are needed.

At lines 51-52, Authors assert that “a three-dimensional culture of MSCs at specific density can potentially generate a dentin/pulp-like structure both in vitro and in vivo”: please add at least one specific reference to this sentence.

Materials and Methods

The paragraph 2.5 should be slightly improved:

  • Please, add the amount of total RNA used to perform the reverse-transcription and the cDNA synthesis kit used.
  • Please, give information about the instrument used to perform gene amplification and software used to perform data analysis.
  • Please, specify the final amplification volume indicating also the volume of cDNA used as template.
  • In the Table 1, please add the amplicon length for each gene.

This reviewer strongly recommend to follow the MIQE guidelines (The MIQE guidelines: minimum information for publication of quantitative real-time PCR experiments. Clin Chem. 2009 Apr;55(4):611-22. doi: 10.1373/clinchem.2008.112797), not only to use the correct nomenclature in the text, but also to enable other investigators to reproduce results.

The use of reference genes as internal controls is the most common method for normalizing cellular mRNA data. However, the reference gene should be stably expressed between treatment groups. Hence, normalization against a single reference gene (e.g., GAPDH), as reported by the Authors,  is not acceptable unless the investigators present clear evidence that confirms its invariant expression under the experimental conditions described. To this regard, has the M-value (the measure of expression stability) of your reference gene been calculated?

Discussion

At lines 248-249, Authors assert that “In accordance with the results from previous researches studying the relative superiorities of Ad-MSCs compared to the MSCs derived from bone marrow and dental tissues”: please add appropriate references to this sentence because this statement is controversial in the scientific literature. In fact, other Authors consider MSCs of pulp origin as the cell source with the highest potential for differentiating into odontoblast-like cell (The efficacy of mesenchymal stem cells to regenerate and repair dental structures, Orthod Craniofac Res 2005, 8, 191–199; Mesenchymal stem cells derived from dental tissues vs. those from other sources: their biology and role in regenerative medicine, J Dent Res 2009, 88, 792–806).

What are the advantages of using adipose derived stem cells instead of induced pluripotent stem cells? For pulp-dentin regeneration approaches, will liposuction vs reprogramming be a real advantage for patients?

At lines 264 – 269, Authors assert that “As previously demonstrated, the mitogen-activated protein kinase (MAPK) and nuclear factor kappa B (NF-κB) pathways are involved in the overexpression of these master regulators and downstream elements and play an important role in osteo/odontogenic inducing potential of MTA on both BM-MSCs and dental stem cells”: please, add appropriate references to these sentences.

Other points to be addressed

In the main text (e.g, page 6 line 194, page 7 line 221) and in figure legends (e.g., figures 3, 4, 5 and 6), when Authors refer to one single concentration of MTA in the DMEM-LG, please use “MTA-conditioded medium” instead of “MTA-conditioned media”.

In the Discussion section, please mention the relative figure number when your results are listed.

Author Response

Thank you for the thorough comments. We believe that by considering these comments, the article has been improved very much.

1- The abstract has been changed to fit the 200-word limit.

2- The introduction part has been elaborated in a way that the comments are answered. The reference has been added as well to address the sentence.

3- All the required numbers and rate have been added to the revised manuscript. Regarding the M value, the authors confirmed that there was no significant variation between the expression of the house keeping gene in different group. This has been added to the revised manuscript as well.

4- Since indicating this sentence can arise controversy, the authors believe that by omitting the dental stem cell from the sentence, it would be more justifiable.

5- Comparison between using mesenchymal stem cells and iPSCs has been elaborated in the discussion section of the revised issue.

6- The appropriate references and addressing to the proper figure have been added.

Round 2

Reviewer 1 Report

General Comments:

  In this manuscript, the author investigated the effects of MTA on the proliferation and differentiation of human adipose derived mesenchymal stem cells (Ad-MSCs) as surrogate candidate cell source in futuristic stem cell-based endodontic therapies.

  I recommend the accept of this revised manuscript.

Reviewer 2 Report

all questions were modified

i reccomend work acceptation in this form